# Creep Damage and Deformation Mechanism of a Directionally Solidified Alloy during Moderate-Temperature Creep

**Jiachun Li [1], Ning Tian [2,\*], Ping Zhang [1], Fang Yu [1], Guoqi Zhao [2] and Ping Zhang [2]**

[1] School of Mechanical Engineering, Guizhou University, Guiyang 550025, China; 18985165581@163.com (J.L.); zhangping_my@163.com (P.Z.); yufang5042021@163.com (F.Y.)
[2] School of Mechanical Engineering, Guizhou University of Engineering Science, Bijie 551700, China; zhaoguoqi1986@163.com (G.Z.); zp0930@126.com (P.Z.)
\* Correspondence: syhgxytn@163.com

**Abstract:** Through creep performance tests, microstructural observations, and contrast analysis of the dislocation configuration, the deformation and damage mechanism of the directionally solidified nickel-based superalloy during creep at moderate temperatures was investigated. The findings suggested that the deformation of the alloy in the late stage of creep at moderate temperatures involved dislocations slipping in the $\gamma$ matrix and shearing into the $\gamma'$ phase. The super-dislocations sheared into the $\gamma'$ phase could either be decomposed to form a <112> super-Shockley incomplete dislocation plus superlattice intrinsic stacking fault (SISF) configuration, or it could slip from the {111} plane to the {100} plane and decompose to form a dislocation configuration of the Kear–Wilsdorf (K-W) lock plus antiphase domain boundary (APB). The configurations of the dislocations could inhibit the slipping and cross-slipping of dislocations to enhance the alloy creep strength, which is thought to be one reason that the alloy displayed good creep resistance. In the late creep stage, the primary/secondary slipping systems were alternately activated, and the interaction of the slipping traces caused micro-holes to appear on the interface of the $\gamma/\gamma'$ phases at the intersection areas of the two slipping systems. The micro-holes gathered and grew to form micro-cracks, which extended along the grain boundary at 45° to the stress axis until creep rupture occurred. These were the damage and fracture characteristics of the alloy in the late stage of creep at moderate temperatures.

**Keywords:** directionally solidified superalloy; microstructure; creep; deformation mechanism; initiation and propagation of crack

## 1. Introduction

In contrast to polycrystalline alloys, the grain boundaries perpendicular to the stress axis within directionally solidified (DS) nickel-based superalloys are removed. Therefore, the alloys exhibit better creep and mechanical properties [1–3]. Although DS alloys have excellent creep resistances, creep damage has been their major failure mode during service. In creep at high temperatures, the alloy microstructure changes significantly. The main changes include $\gamma'$ phase coarsening, dislocations shearing into the $\gamma'$ phase accompanied by the formation and damage of the dislocation network at the $\gamma'/\gamma$ two-phase interface, and the initiation and propagation of cracks along grain boundaries [4–6].

Because blade components in aerospace engines operate in environments ranging from medium temperatures/high stresses to high temperatures/low stresses during service, the blade components experience different operating conditions [7]. Alloys of different compositions exhibit various creep properties, and different deformation mechanisms in different temperature ranges. In particular, the investigation of the movement modes of dislocations under medium-temperature/high-stress conditions has important guiding significance for the development and application of alloys. It is generally accepted that thermal activation causes dislocations to overcome Orowan [8–10] resistance to slipping in the $\gamma$ matrix channel along the <110> direction on the {111} plane [11,12] at the initial

stage of creep under medium-temperature/high-stress conditions. During steady-state creep, the activated dislocations slip in matrix channel and react with each other to form a dislocation network, which can improve the creep resistance of the alloy.

Furthermore, the dislocation shearing into the $\gamma'$ phase is broken down to form the configurations of the superlattice intrinsic stacking fault (SISF) or anti-phase boundary (APB). However, various decomposition models of dislocations have been proposed [13,14]. The dislocations shearing into the $\gamma'$ phase can be broken down to form 1/2<110> partial dislocations [13] during the creep of the CMSX-2 alloy under mid-temperature/low-stress conditions. However, under the same conditions, the dislocations shearing into the $\gamma'$ phase of the DZ4 alloy may be broken down to form 1/3<112> partial dislocations [11]. After the Ni-based alloy has creep up to fracture under 760 °C/800 MPa, the 1/3<112> and 1/6<112> partial dislocations have been placed on two sides of the stacking fault in the $\gamma'$ phase [14]. This suggests that the different alloys show different deformation mechanisms during creep under the same conditions.

During the creep of as-cast nickel-based alloys under high temperatures, microvoids formed during solidification turn into microcracks, and the microcracks can spread along the microvoids and propagate along the $\gamma/\gamma'$ interface perpendicular to the stress axis [15–17] up to fracture. Studies on the creep rupture of DD3 and DD6 single-crystal alloys at high temperatures have shown that the aggregation and nucleation of microvoids may lead to crack generation and propagation. This has been considered to be one main reason that creep fracture occurs during service at high temperatures [18,19]. Research on the high-temperature creep rupture of FGH95 powder nickel-based alloys showed that crack initiation and propagation along the grain boundaries are the creep rupture mechanism of the alloy [20,21].

Although transverse grain boundaries have been eliminated in DS alloys, some longitudinal boundaries remain along the direction parallel to the stress axis [22]. In the process of service, the aerospace engines experience conditions ranging from medium temperature (700–900 °C)/high stress to high temperature (900–1100 °C)/low stress, and so aerospace engine blades must bear various conditions from starting to stabilization. So, it is particularly important to study the medium-temperature creep behaviors of nickel-based superalloys and the deformation mechanism during creep. Although the creep behaviors and deformation mechanisms of DS alloys at high temperatures have been reported [23,24], the damage and fracture behaviors of these alloys during creep at moderate temperatures remain unclear.

Based on this, the creep properties of DS Ni-based alloys were tested under different temperature conditions at 700 MPa. Combined with microstructure observations and contrast analysis of the dislocation configuration, we analyzed the deformation mechanism of medium temperatures creep at different times and examined whether the alloy forms a K-W lock. We also studied the fracture characteristics of the alloy under medium temperature creep and provide a theoretical basis for alloy design and manufacture.

## 2. Experimental Procedure

Bars of a DS nickel-based superalloy with columnar crystal structures along the [001] orientation were made using a vacuum directional solidification furnace. The superalloy chemical composition was Ni-$Cr_{8.68}$-$Co_{9.80}$-$W_{7.08}$-$Mo_{2.12}$-$Al_{5.24}$-$Ti_{0.94}$-$Ta_{3.68}$-$Hf_{1.52}$-$B_{0.012}$-$C_{0.09}$ (mass fraction). The heat-treated alloy test bar was cut into block specimens. After mechanical grinding and polishing, an etching solution of $HNO_3$ + HF + $C_3H_8O_3$ (volume ratio 1:2:3) was used for chemical etching, and scanning electron microscopy (SEM, S-3400N, *HITACHI*) was used to examine the morphology of the alloy in different states.

The bar of the alloy was cut into plate-shaped creep specimens with cross sections of 4.5 mm × 2.5 mm and gauge lengths of 15 mm. After grinding and mechanical polishing, the specimens were put into a creep testing machine (GWT504) to measure the alloy creep curve at various temperatures at 700 MPa. Five specimens were used for creep tests of 780 °C/700 MPa, 790 °C/700 MPa, 800 °C/700 MPa and 790 °C/700 MPa for 50 h and 150 h,

respectively. The microstructure of the alloy in different states was observed by SEM and transmission electron microscopy (TEM, Tecnai-20, ThermoFisher), and the deformation and damage features of the alloy during medium-temperature creep were observed.

## 3. Experimental Findings and Analysis

### 3.1. Microstructure and Creep Characteristics of Alloy

The microstructure along the (001) plane of the alloy after heat treatment is displayed in Figure 1. Because the initial nuclei within the various areas grew opposite to the direction of heat flow during directional solidification, the boundaries of the grains were located along the growing crystals in the direction parallel to the heat flow. In Figure 1, the boundary between grain A and grain B has been marked with a long white line, and some fine carbides were distributed within the grain and boundaries, which are marked with arrows. Figure 1 suggests that having been completely heat treated, the $\gamma'$ phase in alloy exhibited a cubic configuration with a size of about 0.4 μm, while the cuboidal $\gamma'$ phase was regularly placed in the <100> direction. The arrangement direction of the cuboidal $\gamma'$ phase in the grain A is shown by the short white line, while the arrangement direction of the cuboidal $\gamma'$ phase in grain B is shown by the white line on the right. The oriented difference in the cuboidal $\gamma'$ phase between grain A and grain B was about 30°, as indicated in Figure 1. Thus, it was determined that after heat treatment, the microstructure of the DS alloy was made up of the cuboidal $\gamma'$ phase embedded coherently within the $\gamma$ matrix. The grain boundary orientation was the [001] direction, and the misorientation of the cuboidal $\gamma'$ phase appeared in the (001) plane of the different grains. Some fine carbides were distributed within the intragranular regions and boundaries.

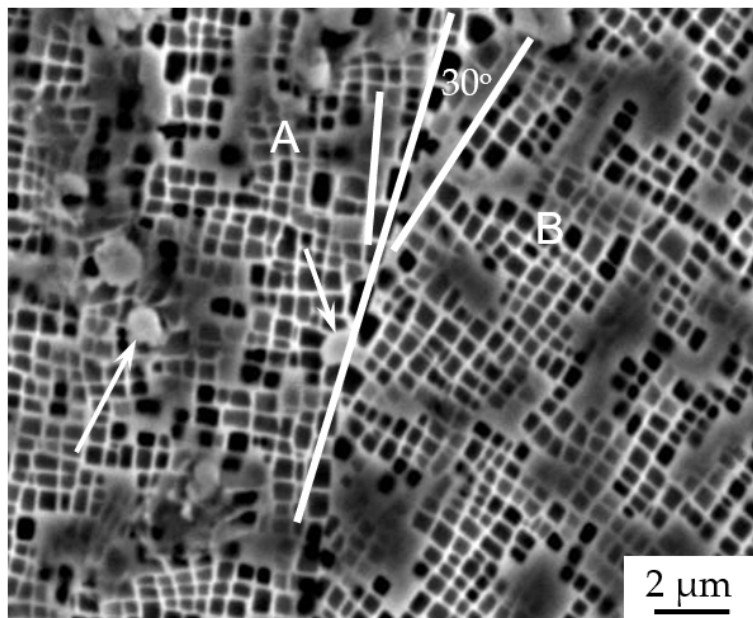

**Figure 1.** Scanning electron microscopy (SEM) morphology on (001) plane of alloy after heat treatment.

The creep curve of the alloy measured at 790 °C/700 MPa as shown in Figure 2. The alloy creep displayed features of primary creep, steady-state creep, and accelerated creep stages. The duration of the primary creep period was shorter, but the strain rate was larger, because a large number of dislocations move in the matrix at the initial creep stage. As the creep time increased, the dislocation density of the alloy increased. Furthermore, its deformation-hardening effect caused the strain rate of the alloy to decrease. Meanwhile, thermal activation caused the dislocations in the alloy to slip and climb, which could release or slow the stress concentration in the local area, and the phenomenon of recovery softening occurred. When the deformation hardening and recovery softening reached a

balance, the creep of the alloy entered a steady-state stage. The steady-state creep lasted for a long time, and the alloy strain rate in the creep period at steady state was measured as 0.024%/h. The duration of the steady-state creep was about 150 h. The alloy life for creep up to fracture was 211 h, and the corresponding alloy strain value was 17%, showing that the alloy has good plasticity compared to similar alloys [25]. The creep curves of the alloy measured at 780 °C/700 MPa and 800 °C/700 MPa are shown in Figure 2. The strain rates under steady-state creep at 780 °C/700 MPa and 800 °C/700 MPa were measured to be 0.008%/h and 0.039%/h, and the alloy's creep life to be 339 h and 133.4 h, respectively.

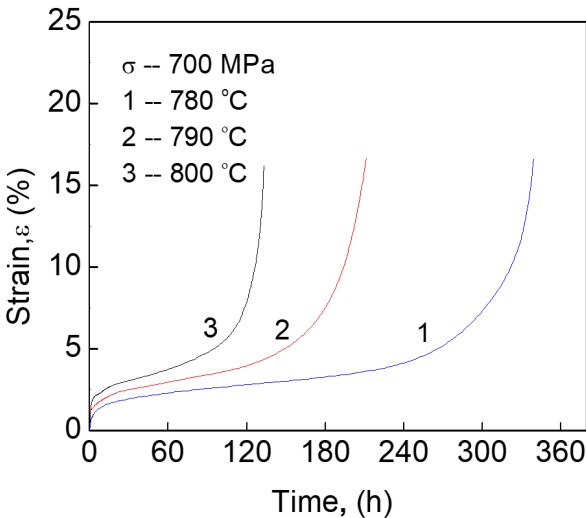

**Figure 2.** Creep curves of the alloy under various conditions.

### 3.2. Deformation Characteristics during Creep

The microstructures within various areas of the alloy after 211 h of creep up to fracture with an applied stress of 700 MPa at 790 °C are shown in Figure 3. A schematic in the monitoring areas within the sample is displayed in Figure 3a, which shows the various evolution characteristics of the sample within various areas due to their different stress states. Thus, the alloy deformation extent of various regions is assessed based on microstructural evolution characteristics in different regions.

The minimum stress area of the specimen is marked with the letter A in Figure 3a, and the microstructure within the area is displayed in Figure 3b. No changes in the morphology and size of the $\gamma'$ phase were evident within this region. The $\gamma'$ phase within this region retained the same cubic configuration with a size of about 0.4 μm. This was due to the lower elemental diffusion rate during creep at lower temperatures. The microstructure of the specimen in region B with a smaller tensile stress is shown in Figure 3c. No morphology changes in the $\gamma'$ phase were evident at smaller strains, and much more of the $\gamma'$ phase in the region retained a cubic configuration, but the width of the $\gamma$ matrix channel increased slightly. Microstructures in areas C and D are displayed in Figure 3d,e, respectively. The deformation extent increased as the distance from the fracture decreased, and necking occurred near the fracture area, which reduced the effective area of the load. Therefore, as the effective applied stress increased, the degree of $\gamma'$ phase distortion increased, the corners of the cubic $\gamma'$ phase were rounded, and clear kink characteristics were evident in the $\gamma'/\gamma$ two-phase interface area.

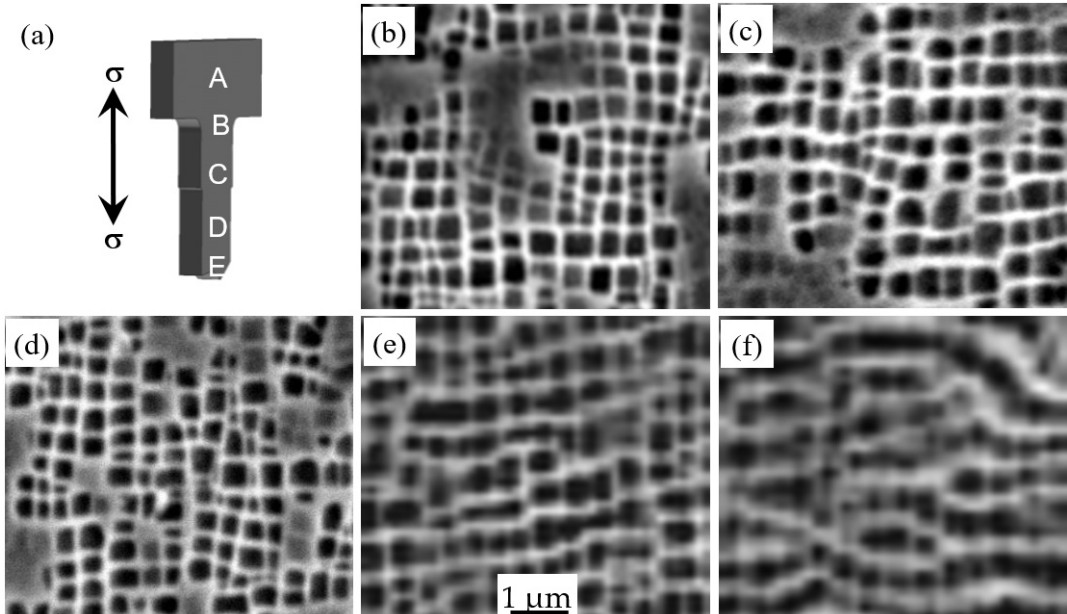

**Figure 3.** Microstructure within various areas of nickel-based superalloy after 211 h of creep up to fracture with an applied stress of 700 MPa at 790 °C. (**a**) Schematic diagram of monitored positions of the specimen, (**b**–**f**) SEM morphologies in regions A, B, C, D, and E, respectively.

The alloy deformation features in the $\gamma/\gamma'$ phases after creep for various times at 790 °C/700 MPa are shown in Figure 4. The alloy deformation features after 50 h of creep are displayed in Figure 4a, and the applied stress direction has been marked with an arrow. This suggests that the $\gamma'$ phase in the alloy retained the cubic configuration. A few dislocations shearing into the $\gamma'$ phase were detected due to the smaller strain occurring within the area, while a large number of (a/2)<110> dislocations slipped and cross-slipped in the narrow matrix channels. The regular dislocation networks were distributed in the interfaces of the $\gamma/\gamma'$ phases, as shown by the white square of Figure 4a. The magnified morphology in the square area is displayed in the upper right side of Figure 4a, and the quadrilateral dislocation networks are marked with arrows. The regular hexagonal dislocation network formed during creep could release the lattice mismatch stress of the $\gamma/\gamma'$ phases.

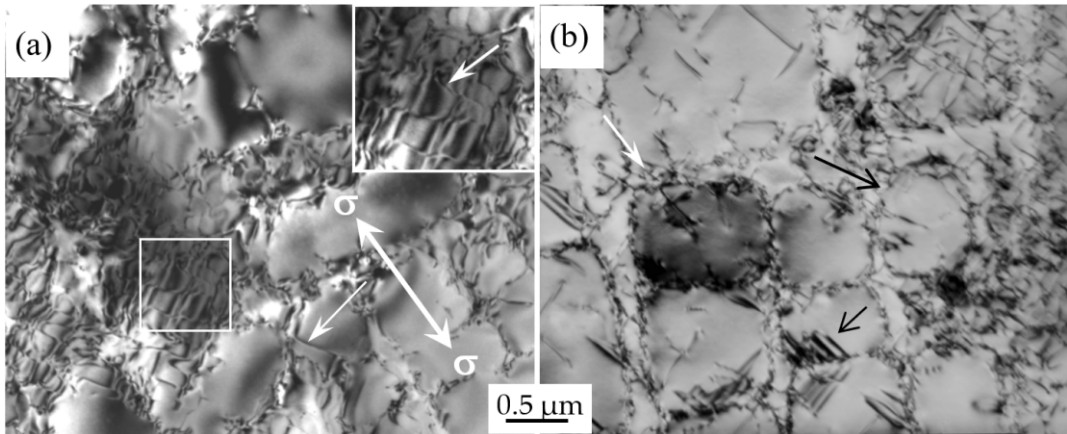

**Figure 4.** Alloy morphology after creep for various times at 790 °C/700 MPa. (**a**) Creep for 50 h, (**b**) Creep for 211 h up to fracture.

When the active dislocations in the matrix moved to the interface and reacted with the dislocation network during creep, the original movement direction of the dislocations could be altered, the climbing of dislocations could be promoted, and the phenomenon of recovery softening occurred. Therefore, the dislocation network could coordinate the deformation hardening and recovery softening phenomena caused by dislocation stacking during creep, which is beneficial for improving the creep resistance of the alloy. The fact that only a small number of dislocations sheared into the $\gamma'$ phase during the steady-state creep of the alloy is closely related to the dislocation network situated in the $\gamma/\gamma'$ phase interface. In addition, a few dislocations shearing into the $\gamma'$ phase displayed a 90° kink feature, which is marked with an arrow within Figure 4a. This was due to the dislocation cross-slipping on the {111} plane.

The alloy deformation characteristics after 211 h of creep at 790 °C/700 MPa until fracture are displayed in Figure 4b. These characteristics suggest that the $\gamma'$ phase within the alloy changed into a spherical configuration from the original regular cubic $\gamma'$ phase, which is marked by the long black arrow in Figure 4b. This indicated that elemental diffusion occurred during alloy creep. Compared to steady-state creep, the number of dislocations shearing into the $\gamma'$ phase within the late creep increased significantly. The alloy strain increased as the creep continued, which resulted in the activated dislocations accumulating during creep at the $\gamma/\gamma'$ phase interfaces to form dislocation tangles, as marked with a white arrow in Figure 4b.

The dislocation pile-up of the interface during creep could generate stress concentration, while the dislocation networks of the interfaces could be damaged when the stress value in the stress concentration region was greater than the yield strength of the $\gamma'$ phase. Therefore, the slipping dislocations within the $\gamma$ matrix sheared into the $\gamma'$ phase in the damaged region of the networks. The dislocations that sheared into the $\gamma'$ phase were broken down to form a dislocation configuration of incomplete dislocations plus stacking faults, wherein one of the partials was situated in the interface of the $\gamma/\gamma'$ phases, and another was located within the $\gamma$ phase, as shown by the short black arrow in Figure 4b.

The alloy deformation features within the area close to the boundary after 211 h of creep up to fracture at 790 °C/700 MPa are displayed in Figure 5. A few fine carbide particles were present within the boundary region, as shown by the arrow. A large number of dislocations were piled up within the area close to the boundary to form dislocation tangles. This suggested that boundaries and carbides impeded the dislocation motion during creep. During creep, when the movement of the active dislocations in the alloy matrix was blocked by the grain boundary, dislocation pile-up could occur near the grain boundary, as shown in the white square area in Figure 5. According to this analysis, the strain value and dislocation density of the alloy increased as the creep progressed. This could cause a strain hardening effect, hindering dislocation motion. The resistance ($\tau_{dis}$) of the dislocation motion from the dislocation pile-up can be expressed as follows [20]:

$$\tau_{dis} = \alpha \mu b \sqrt{\rho} \tag{1}$$

where $\alpha$ is a constant of about 0.1, $\mu$ is the shear modulus of the alloy, b is the Burgers vector of the dislocation, and $\rho$ is the dislocation density in the piled-up region. This suggests that the dislocation motion resistance increased with the dislocation density. Owing to the fact that the boundaries in the alloy had the effect of impeding the dislocation motion during creep, the grain boundary improved the alloy creep resistance at medium temperatures. However, the alloy strain and dislocation density increased with the creep. This could cause stress concentration. The dislocations in the $\gamma$ matrix sheared into the $\gamma'$ phase in the damaged regions of the networks when the stress concentration value exceeded the yield strength of the $\gamma'$ phase, causing crack generation and propagation in the interface of the $\gamma'/\gamma$ phases until the creep rupture of the alloy.

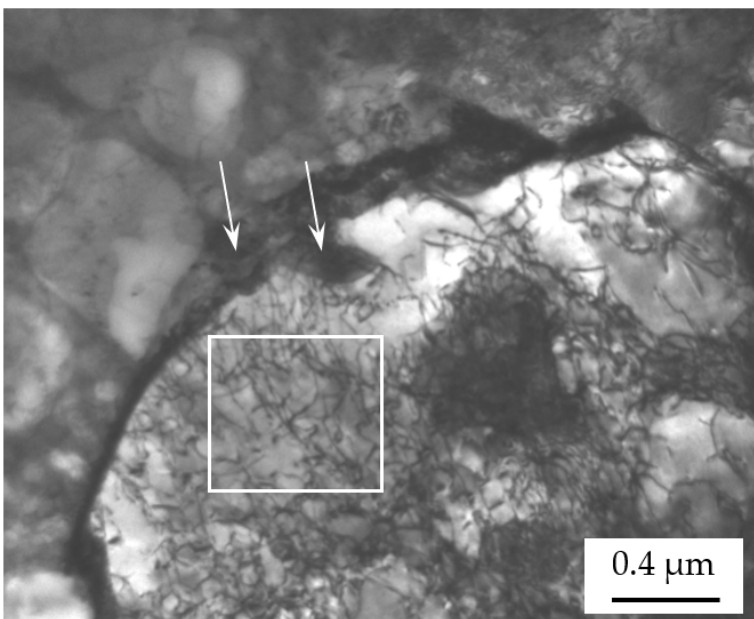

**Figure 5.** Alloy microstructure in the region near the grain boundary after 211 h of creep up to rupture.

The dislocation configurations in the $\gamma'$ phase after 211 h of creep up to fracture at 790 °C/700 MPa are displayed in Figure 6. This suggests that the dislocation had sheared into the $\gamma'$ phase in the late creep stage, while the dislocations shearing into the $\gamma'$ phase could be broken down to form the configuration of the partial dislocation plus stacking faults. The partial dislocations on both sides of the stacking fault are indicated by the letters H and J, and the super dislocations sheared into the $\gamma'$ phase are indicated by the letter K.

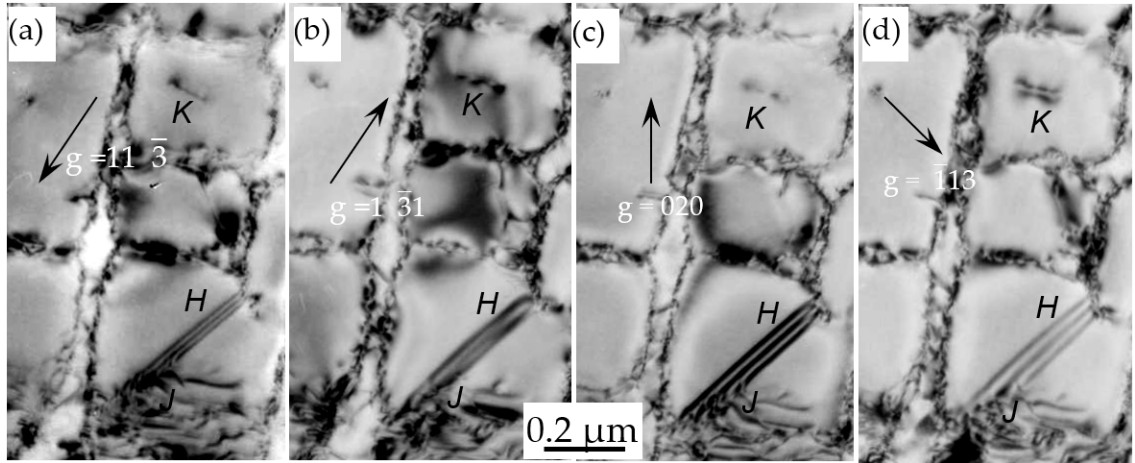

**Figure 6.** Dislocation configuration within $\gamma'$ phase of alloy during creep up to fracture under 790 °C/700 MPa. (**a**) g = $11\bar{3}$, (**b**) g = $1\bar{3}1$, (**c**) g = 020, and (**d**) g = $\bar{1}13$.

As shown in Figure 6c, with the diffraction vector g = 020, the contrast of partial dislocations H and K disappeared. The contrast of partial dislocation H is shown in Figure 6b. For g = $11\bar{3}$ and g = $\bar{1}13$, the partial dislocation H showed contrast, as shown in Figure 6a,d. Based on the invisibility criterion of dislocations, $\boldsymbol{g}\cdot\boldsymbol{b} = 0$ and $\boldsymbol{g}\cdot\boldsymbol{b} = \pm(2/3)$, it was determined that dislocation H was a super-Shockley partial dislocation with a Burgers vector of $\boldsymbol{b}_H = (1/3)[112]$. When the diffraction vectors were g = $11\bar{3}$ and g = $\bar{1}13$, the contrast of dislocation J disappeared, as shown in Figure 6a,d. For g = $1\bar{3}1$, the dislocation J showed contrast, as shown in Figure 6b. Based on the invisibility criterion of dislocations, $\boldsymbol{g}\cdot\boldsymbol{b} = 0$ and $\boldsymbol{g}\cdot\boldsymbol{b} = \pm(2/3)$, it was determined that dislocation J was a super-Shockley

partial dislocation with a Burgers vector of $b_J = (1/6)[2\bar{1}1]$. Therefore, it was determined that the <110> dislocations sheared into the $\gamma'$ phase and were broken down, forming a dislocation configuration of two super-Shockley incomplete dislocations + stacking faults (SISF). Because $b_J \times \mu_J = (11\bar{1})$, it was determined that the dislocation decomposed on the $(11\bar{1})$ plane, and the reaction formula is as follows:

$$[101] \rightarrow (1/3)[2\bar{1}1]_J + (SISF)_{(11\bar{1})} + (1/3)[112]_H. \tag{2}$$

In addition, the super-dislocation shearing into the $\gamma'$ phase has been labeled with the letter K in Figure 6. Under the diffraction vector g = $11\bar{3}$ and g = $1\bar{3}1$, the super-dislocation K in the $\gamma'$ phase showed contrast, as shown in Figure 6a,b. For the diffraction vector g = $\bar{1}13$, the super-dislocation K in the $\gamma'$ phase showed double line contrast, as shown in Figure 6d. When the diffraction vector was g = 020, the super-dislocation K contrast disappeared, as shown in Figure 6c. Based on the invisibility criterion of dislocations, the Burgers vector of the dislocation K was determined to be $b_K = [101]$. The line vector of the dislocation K was $\mu_K = 200$, and thus the slipping plane of the dislocation K was identified as the (001) plane because $b_K \times \mu_K = (001)$.

The $\gamma'$ and $\gamma$ phases in the alloy had FCC (face-centered cubic) and Ll$_2$ structures, respectively, and the slipping during creep occurred easily along the {111} plane. The activated dislocations within the alloy during creep slipped within the $\gamma$ matrix first and then sheared in to $\gamma'$ phase along the {111} plane. As the creep progressed, the super-dislocation K shearing into the $\gamma'$ phase cross-slipped from the {111} to the (001) plane, and it broke down in the (001) plane, forming Kear–Wilsdorf (K-W) locks with non-planar core structures. These K-W configuration locks could effectively prevent the dislocation from slipping and cross-slipping on the {111} plane, improving the alloy creep resistance. The decomposition reaction of the dislocations can be expressed as follows:

$$[101]_K \rightarrow (1/2)[101]_K + (APB)_{(100)} + (1/2)[101]_K. \tag{3}$$

### 3.3. Creep Damage of Alloy

After creeping at 790 °C/700 MPa for different times, the sample surface morphologies are displayed in Figure 7, and the applied stress direction has been marked with arrows. The sample surface morphology after 150 h of creep is shown in Figure 7a, and the alloy strain value at this time was about 5.5%. Due to a twisted boundary appearing in Figure 7a, although the $\gamma'$ phase in the alloy retained a cubic morphology, holes had formed within the boundary areas, as indicated by the arrows in Figure 7a.

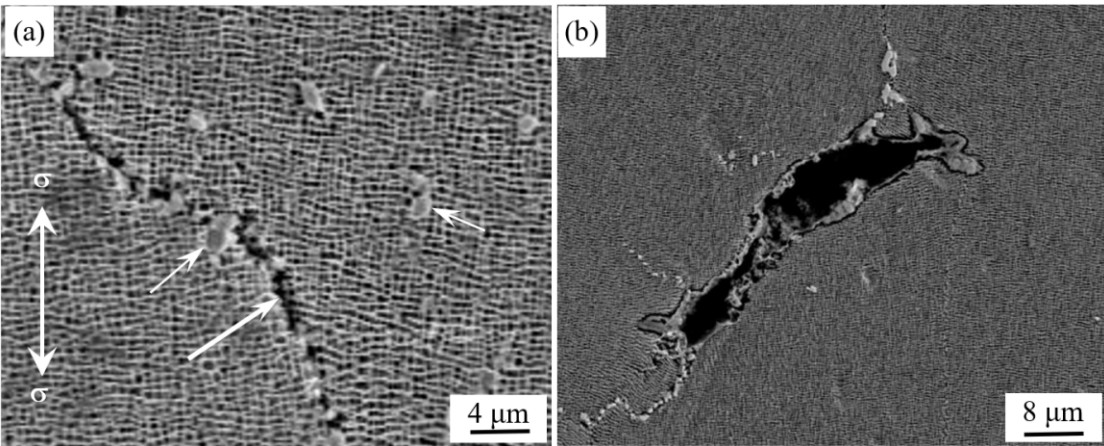

**Figure 7.** Morphology of the grain boundary after different creep times at 790 °C/700 MPa: (**a**) 150 h and (**b**) 211 h up to fracture.

During creep, the denser dislocations that accumulated within the area close to the boundaries could cause stress concentration. Once the value of the stress concentration exceeded the bonding strength of the boundary, cracks were generated first in the boundary area. The stress concentration was released at the moment the crack initiated in the stress concentration region, which led to the smooth creep progression. Analysis showed that when the applied load generated shear stress along the grain boundary during creep, the shear stress could promote crack propagation along the grain boundary. In addition, the grain boundary at the angle of about 45° to the stress axis bears the maximum shear stress of the applied load, which could promote the appearance of a large crack at the grain boundary at an angle of 45° to the stress axis. Moreover, a few fine carbide particles were distributed within the grain boundary, as indicated by short arrows in Figure 7a. These fine carbides could effectively prevent the grain boundaries from sliding and hinder crack propagation.

During the late creep stage, hole accumulation and growth occurred within the grain boundary, promoting the initiation of cracks. As the creep progressed, crack propagation occurred along the grain boundary as the alloy strain increased. Different cross-section propagation cracks were connected to each other, which reduced the effective area of the alloy and increased the effective stress. This could further increase the strain of the alloy. When the opening displacement of the crack tip increased to the critical value, the unstable propagation of the crack occurred until the creep rupture of the alloy. This was the alloy fracture mechanism during the late creep stage at moderate temperatures. During the late creep at 790 °C/700 MPa, the initiation and propagation of cracks occurred easily in the grain boundary 45° to the stress axis, and the larger cracks formed within this area, as shown in Figure 7b. The fact that the cracks were easily generated and propagated along the grain boundary during creep meant that the grain boundaries in the alloy remained the weak link of the creep strength.

Creep tests at 790 °C/700 MPa were conducted for different times, and the surface morphologies of other areas of the sample are displayed in Figure 8. The direction of the applied stress and slip system traces in the specimen are marked with arrows. The sample surface morphology after 150 h of creep is shown in Figure 8a. Slip system traces with a double orientation at a 45° angle to the stress axis appeared, which was caused by the maximum decomposed shear stress of the applied load, as indicated by the lines in Figure 8a. During the late creep stage, the primary or secondary sliding dislocations were alternately activated, and the $\gamma/\gamma'$ phase was distorted as the creep progressed. In particular, the two sets of slipping traces intersected each other to make holes appear in this region. The accumulation and growth of micro-holes occurred as the creep progressed, which promoted the initiation of cracks, as marked with arrows in Figure 8a.

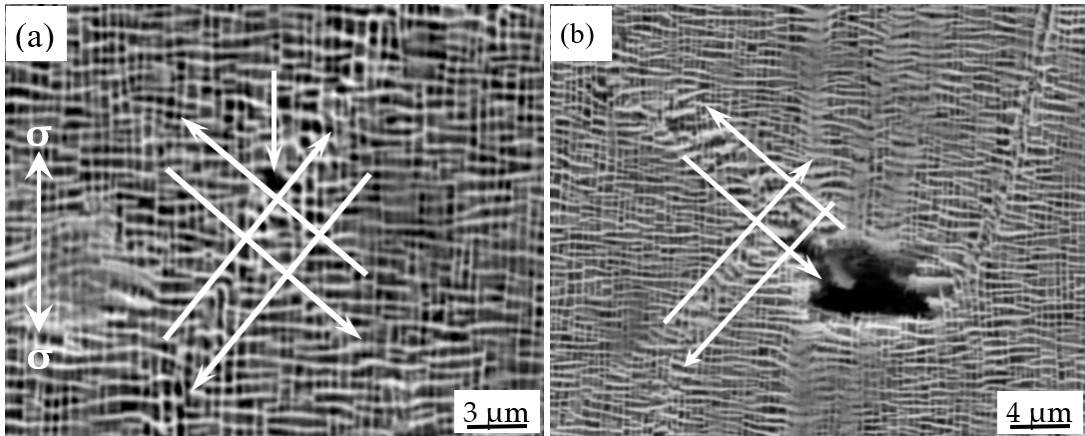

**Figure 8.** Surface morphology of the sample after various creep times under 790 °C/700 MPa. (**a**) Creep for 150 h, (**b**) Creep for 211 h up to fracture.

The crack initiation and propagation morphology of the sample near the fracture area after 211 h of creep up to fracture at 790 °C/700 MPa is shown in Figure 8b. In the later stage of creep, the alternating movement of the primary and secondary slip systems caused plastic deformation of the sample along the direction perpendicular to the stress axis. In particular, the stress concentration in the delivery area of the slip trace promoted the formation of microcracks.

As the creep progressed, the alternate activation of the primary and secondary slip systems caused multiple micro-cracks to expand and connect to each other, forming large cracks, as shown by the arrows in Figure 8b. When multiple large cracks with different cross-sections extended and connected to each other, the effective area of the alloy that could bear a load was reduced, the effective load stress increased, and the crack entered the unstable propagation stage until the creep fracture of the alloy occurred, which was the damage and fracture mechanism of the alloy in the later stage of creep.

## 4. Discussion

### 4.1. Creep Strength and Factors Affecting Alloy

The microstructure of the DS alloy was the cubic $\gamma'$ phase embedded in the $\gamma$ matrix in a coherent manner, with grain boundaries along the [001] orientation, and granular carbides within and at the grain boundaries, as shown in Figure 1. The cubic $\gamma'$ phase precipitated in the grains and the granular carbide precipitated on the grain boundaries were the strengthening phases of the alloy. During the initial and steady-state creep, the dislocations only moved in the $\gamma$ matrix, and the granular $\gamma'$ phases and the carbides could hinder the movement of dislocations.

Figures 4 and 5 show that there were different microstructural characteristics and dislocation numbers at the grain boundaries and in the grains after creep fracture. In particular, the grain boundary direction was parallel to the stress axis, and the $\gamma$ and $\gamma'$ phases together with carbide particles were homogeneously located within the grains. Thus, various creep resistances appeared within different areas of the alloy. Therefore, in the microstructure analysis, the alloy can be regarded as being composed of grain boundaries and intragranular regions, and the intragranular regions can be divided into (1) $\gamma$ matrix softened regions, (2) $\gamma'$ strengthening phase regions, and (3) carbide particle strengthening regions, which are referred to as the alloy substructure. During creep at high temperatures, the redistribution of applied stress occurred when the sample was deformed at the same strain rate. $\sigma$ is an external applied stress, $\sigma_G$ is the local stress within the grains, $\sigma_{GB}$ is the local stress of the boundaries, $d_{GB}$ is the width of the grain boundary region, $d_\gamma$ is the width of $\gamma$ phase, $d_{\gamma'}$ is the width of $\gamma'$ phase, and the width of the columnar crystals in the lateral direction is defined as $d$. Therefore, the strength of the alloy may be expressed as follows:

$$\sigma = \sigma_G \frac{d - d_{GB}}{d} + \sigma_{GB} \frac{d_{GB}}{d} \tag{4}$$

Because $d_{GB} << d$, this equation may be expressed as follows:

$$\sigma = \sigma_G + \sigma_{GB} \frac{d_{GB}}{d} \tag{5}$$

Because the $\gamma/\gamma'$ phases and carbide particles were distributed within the grain regions, the local intragranular stress ($\sigma_G$) can be expressed as follows:

$$\sigma_G = \sigma_\gamma + \sigma_P + \sigma_{\gamma'} \tag{6}$$

where $\sigma_\gamma$ is the local stress of the $\gamma$ matrix phase, $\sigma_P$ is the local stress of the carbide particles, and $\sigma_{\gamma'}$ is the local stress of the $\gamma'$ phase. When dislocations moved to the grain boundaries, the phases and carbides were hindered, which could cause dislocation plugging. As shown in Figure 5, the deformation hardening effect could improve the creep

resistance of the alloy [26,27]. Therefore, the local stress ($\sigma_{GB}$) of the grain boundary can be expressed by the dislocation plugging stress as follows [28]:

$$\sigma_{GB} = m\sigma\left(\frac{L}{r}\right)^{1/2} \tag{7}$$

Here, $m$ is the density of the $\gamma'$ phase and carbide particles, and $r$ is the size of the $\gamma'$ phase and carbide particles. The size of the $\gamma'$ phase is close to the size of the carbides and grain boundaries, so $r \approx d_{GB} \approx d_{\gamma'}$. $L$ is the distance of dislocation slipping. It is assumed that $L = 8d_\gamma = 2d_{\gamma'}$, and the $\gamma'$ phase, carbides, and grain boundaries are regarded as the substructure of the alloy. Because both the cubic $\gamma'$ phase and carbide particles within the grain boundaries could hinder the dislocation motion and restrain the slipping of grain boundaries during creep, the local stress ($\sigma_{GB}$) of the grain boundaries can be expressed as follows:

$$\sigma_{GB} = m(\sigma - \sigma_P - \sigma_{\gamma'})\left(\frac{L}{r}\right)^{1/2} = m(\sigma - \sigma_P - \sigma_{\gamma'})\left(\frac{2d_{\gamma'}}{d_{GB}}\right)^{1/2} \tag{8}$$

The relationship between the applied stress, strengthening phase, and grain boundary substructure size ($d_{GB}$) is as follows:

$$d_{GB} = \frac{KGb}{\sigma} \tag{9}$$

where $K$ is a constant related to the material, $G$ is shear modulus, and $b$ is the Burgers vector of the dislocation. After $d_{GB}$ in Equation (9) is substituted into Equation (8), the following equation is obtained:

$$\sigma_{GB} = m(\sigma - \sigma_p - \sigma_{r'})\left(\frac{2\sigma \cdot d_{r'}}{K \cdot G \cdot b}\right)^{\frac{1}{2}} \tag{10}$$

Equations (6) and (10) are substituted into Equation (4). After reorganizing, the following is obtained:

$$\sigma = \sigma_\gamma + \sigma_P + \sigma_{\gamma'} + m(\sigma - \sigma_p - \sigma_{r'})\frac{(2d_{r'}K \cdot G \cdot b)^{1/2}}{d \cdot \sigma^{1/2}} \tag{11}$$

The following is defined:

$$\sigma_{bo} = \frac{m(2d_{r'}K_1 G \cdot b \cdot \sigma)^{\frac{1}{2}}}{d} \tag{12}$$

where $\sigma_{bo}$ is the strengthening capacity caused by the precipitates in the boundary. Equation (12) shows that $\sigma_{bo}$ is inversely proportional to the grain size and directly proportional to the density ($m$) of the $\gamma'$ phase and carbides. Equation (11) can be rewritten as follows:

$$\sigma = \sigma_r + \sigma_p + \sigma_{r'} + \sigma_{bo}\left(1 - \frac{\sigma_p}{\sigma} - \frac{\sigma_{r'}}{\sigma}\right) = \sigma_r + \sigma_p + \sigma_{r'} + \sigma_{BO} \tag{13}$$

where

$$\sigma_{BO} = \sigma_{bo}\left(1 - \frac{\sigma_p}{\sigma} - \frac{\sigma_{r'}}{\sigma}\right) \tag{14}$$

$\sigma_{BO}$ is the resistance to the grain boundary sliding when the strengthening phase is located in the grain boundary. When $\sigma_{BO} = \sigma_{bo}$, no precipitated phase was located in the grain boundary. The value of $\sigma_{BO}$ decreased as the number of precipitates in the grain boundary increased. At this point, the grain boundary strength was dominated by the strengthening effect of the precipitated phase. Because the $\gamma$ matrix within the grains was a softened phase, it was easy to deform and activate dislocations during creep, but the cubic $\gamma'$ phase and carbide particles could efficiently impede the dislocation motion as well as restrict the grain boundary sliding. Therefore, the constitutive equation for the local stress

($\sigma_\gamma$) of the intragranular matrix promoted the creep of the alloy during steady-state creep, which can be expressed as follows [29]:

$$\dot{\varepsilon} = A(\frac{\sigma - \sigma_p - \sigma_{r'} - \sigma_{BO}}{E})^n exp(-\frac{Q}{RT}) \tag{15}$$

where $\dot{\varepsilon}$ is the strain rate, $A$ is a constant, $E$ is the elastic modulus of the material, $Q$ is the apparent activation energy of creep, $n$ is the apparent stress exponent, $R$ is the gas constant, and $T$ is the absolute temperature. Equation (15) shows that the slipping resistance of the boundary increased as the quantity of the precipitated strengthening phase increased. This decreased the alloy strain rate during steady-state creep and increased the creep resistance of the alloy. The above analysis is consistent with the experimental results.

### 4.2. Theoretical Analysis of Crack Generated and Propagated along Grain Boundary

Although transverse grain boundaries perpendicular to the stress axis could be eliminated by the directional solidification technique, the grain boundaries parallel to the stress axis remained within the inter-dendrite region, as shown in Figure 1, and the grain boundaries are still the weaker link of the alloy during creep. During creep, the dislocations were activated first and then slipped in the $\gamma$ matrix. They could pile up within the region close to boundary, causing stress concentration when the activated dislocations in the $\gamma$ matrix were impeded through the grain boundary, as shown in Figure 5. The stress concentration value caused by the piling up of dislocations in the boundary region is expressed as follows [30]:

$$\sigma = n\sigma_0 \tag{16}$$

where $\sigma$ is the value of the stress concentration of the grain boundary and $\sigma_0$ is the applied stress. Equation (16) suggests that during creep, the stress concentration value of the grain boundary region was about $n$ times the applied stress. Because the grain boundary in the alloy was still the weaker link of the creep strength, the crack generation and propagation occurred first within the grain boundary region during creep, as shown in Figure 7. In addition, some grain boundaries along the direction parallel or inclined to the stress axis were still kept within the alloy. Because the angles between the grain boundaries and the stress axis were related to stress state of the grain boundary, the crack initiation and propagation of different grain boundaries in the alloy had different characteristics.

The critical shear stress of the boundary at various angles to the stress axis is expressed as follows: $\tau = \sigma_0/2\pi(2\sin2\alpha)$, where $\tau$ is the shearing stress of the grain boundary at different angles to the stress axis, $\sigma_0$ is the applied stress, and $\alpha$ is the angle between the outer normal of the cross section and the outer normal of the oblique section. This indicates that $\sin2\alpha$ had a maximum value when $\alpha = 45°$. Therefore, the probability of the crack being generated and propagating along the grain boundaries parallel to the stress axis was smaller. The grain boundaries with an angle about $45°$ to the stress axis experienced the largest shearing stress during creep, so the crack generation and propagation occurred simply within the region of the inclined boundary, which is shown in Figure 9a,b.

The schematic diagrams of the cracks that initiated and propagated along the boundary are shown in Figure 9, and the applied stress direction is marked with arrows. The boundary at about a $45°$ angle to the stress axis is marked with black lines. Because the boundaries that were about $45°$ to the stress axis experienced the largest shearing stress during creep, a hole appeared first in this region, as shown by the arrow in Figure 9a. As the creep progressed, vacancies formed within the boundaries and diffused along the boundary into a hole. The hole grew as it absorbed the vacancies, which caused adjacent holes to become connected, forming a larger crack. The aggregation and growth of the holes in the boundary occurred to form the micro-cracks, which is shown with a fine arrow in Figure 9b. In the later creep stage, the micro-crack further propagated along the boundary to form a larger crack, as shown by the larger arrow in Figure 9b. As the creep progressed, cracks with different cross-sections further expanded, causing the cracks to connect to each

other until the creep fracture of the alloy occurred, which was the damage and fracture mechanism of the alloy in the later stage of creep. The above analysis was consistent with the results in Figure 7.

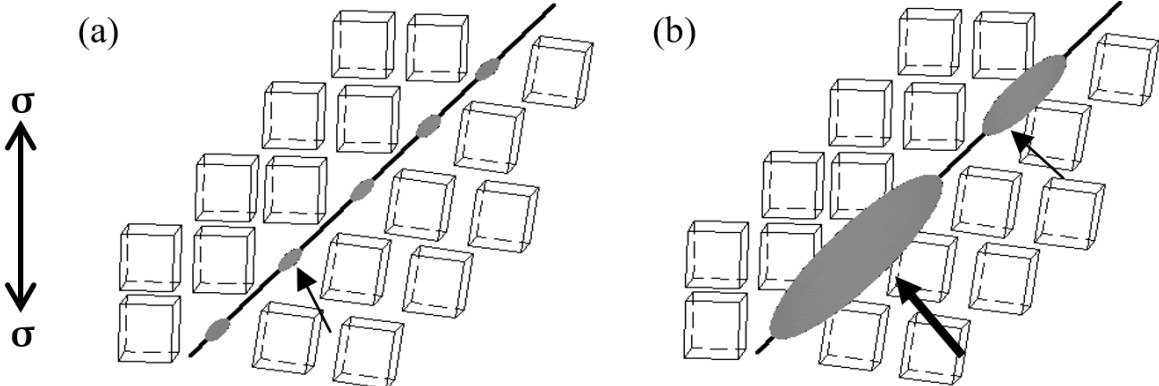

**Figure 9.** Schematic diagrams of the cracks generated and propagated along boundaries: (**a**) 150 h of creep and (**b**) 211 h of creep up to fracture.

### 4.3. Influence of Deformation Mechanism on Creep Resistance

The $\gamma'$ phase (Ni3Al) is a long range ordered intermetallic compound with a LI$_2$ crystal structure, and is the main strengthening phase of nickel-based single crystal alloys [28]. The dislocations cut into the $\gamma'$ phase decompose during the creep. The schematic diagram of the dislocations cutting into the $\gamma'$ phase decomposing on the {111} plane and the {100} plane is shown in Figure 10. The direction of the Burgers vector of the dislocation is shown by the arrow in the Figure 10, and the slip direction of the dislocation is perpendicular to the Burgers vector. Figure 10a shows the super dislocation cut into the $\gamma'$ phase along the {111} plane and decomposed according to Equation (2) to form the Shockley incomplete dislocations K, G and SISF formed in Figure 6.

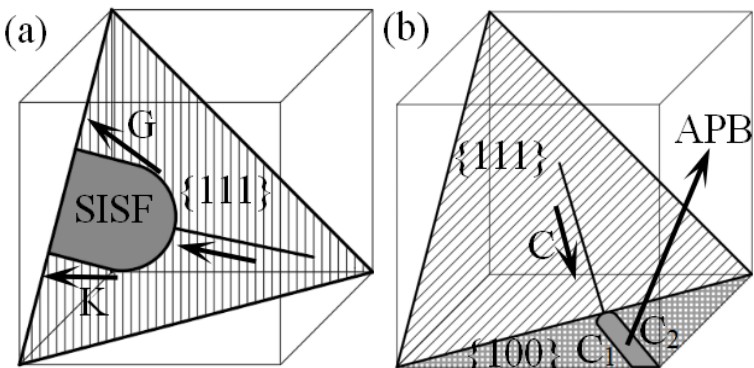

**Figure 10.** Schematic diagram of dislocations shearing into $\gamma'$ phase and decomposing to SISF or APB: (**a**) Dislocations decompose on the {111} plane to form SISF and (**b**) Dislocations decompose on the {111} plane to APB.

The analysis suggests that due to the low formation energy of the {111} plane superlattice intrinsic stacking fault (SISF), when the dislocation shears along the {111} plane into the $\gamma'$ phase, it is easy to decompose on the {111} plane and form (a/3)<112>Incomplete dislocation+SISF+(a/6)<112> stable structures of incomplete dislocation. The larger width of SISF makes it difficult for the Shockley incomplete dislocations on both sides of the stacking fault to cluster, so the slip and cross slip cannot occur, which can improve the creep resistance of the alloy. Moreover, due to the low energy required for dislocation decompo-

sition to form APB at the {100} plane, the structure is relatively stable [30]. Therefore, when the dislocation cross slides to the {100} plane to form a K-W lock, the dislocation on the {100} plane can undergo dislocation decomposition, forming two (A/2) <110> incomplete dislocation + APB dislocation structures as shown in Figure 10b, which is easy to retain until creep fracture occurs. The K-W lock is a dislocation configuration with a non-planar core structure, which can inhibit the slip and cross slip of dislocations. Therefore, this dislocation configuration can also improve the creep resistance of the alloy [12].

**5. Conclusions**

(1) The deformation of an alloy in the period of steady-state creep at moderate temperatures involved dislocations slipping in the $\gamma$ matrix and a regular interfacial dislocation network forming between cubic $\gamma'/\gamma$ phases. The deformation of the alloy in the late creep stage involved super-dislocations shearing into the $\gamma'$ phase, wherein the super-dislocations that sheared into $\gamma'$ phase could be broken down to form <112> super-Shockley partial dislocations. Super-lattice intrinsic stacking faults (SISF) existed between two partial dislocations.

(2) During creep at 700 °C/700 MPa, the super-dislocations shearing into the $\gamma'$ phase could cross-slip from the {111} plane to the {100} plane and decompose in the {100} plane to form a Kear–Wilsdorf (K-W) lock with a non-planar core structure plus an anti-phase boundary (APB). The K-W dislocation lock could inhibit the slipping and cross-slipping of dislocations to enhance the creep strength of the alloy, which is thought to be the reason that the alloy displayed good creep resistance.

(3) In the late creep stage, the primary and secondary slipping systems were alternately activated, and the interaction of the slipping traces caused micro-holes to appear on the interface of the cubic $\gamma/\gamma'$ phases within the intersection regions of the slipping systems. The accumulation and growth of micro-holes occurred as the creep progressed, forming micro-cracks that propagated along the boundary 45° to the stress axis until the creep fractured. This is thought to be the damage and fracture mechanism of the alloy in the late creep stage at moderate temperatures.

**Author Contributions:** Conceptualization, N.T.; methodology, N.T. and J.L.; software, G.Z.; validation, N.T., J.L.; investigation, P.Z.1; data curation, F.Y., P.Z.2; writing—original draft preparation, N.T., J.L.; writing—review and editing, N.T., J.L.; supervision, J.L.; project administration, N.T., J.L.; funding acquisition, N.T., J.L., P.Z.1, G.Z. All authors have read and agreed to the published version of the manuscript.

**Funding:** This research was funded by The Science and Technology Foundation Project of Guizhou province, grant number qiankehejichu [2020]1Y198, qiankehezhicheng[2019]2870, qiankehezhicheng [2018]1055 and The Science and Technology Project of Bijie City, grant number bikehezi[2019]2 and The Characteristic Key Laboratory of University of Guizhou province, grant number qianjiaoheKYzi[2019]053 and The Youth Science and Technology Growth Project of Guizhou Province, grant number qianjiaoheKYzi[2020]145 and The Scientific Research Fund of Guizhou University of Engineering Science, grant number G2017001 and The Guizhou Provincial Education Department Supporting Program, grant number qianJiaoheKY[2019]159.

**Informed Consent Statement:** Not applicable.

**Data Availability Statement:** The raw/processed data required to reproduce these findings cannot be shared at this time as the data also forms part of an ongoing study.

**Acknowledgments:** The authors are cordially thankful to the honorable reviewers for their constructive comments to improve the quality of the paper.

**Conflicts of Interest:** The authors declare no conflict of interest.

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
