# Peer review of "Creep Damage and Deformation Mechanism of a Directionally Solidified Alloy during Moderate-Temperature Creep"

_crystals, doi:10.3390/cryst11060646_

Round 1

Reviewer 1 Report

Manuscript: crystals-1188168

The manuscript titled 'Creep Damage and Deformation Mechanism of a Directionally Solidified Alloy During Moderate-Temperature Creep' by authors Li Jiachun et al present results on the the deformation and damage mechanism of the directionally solidified nickel-based superalloy during creep at moderate temperatures. I have following observations:

1// Figure 2 present alloy creep curve at 790°C/700 MPa. What is the reason to only choose 211 h for measurements and what happens after that measurement time (it will be good to extend the curve). Please also discuss reasons for primary creep and accelrated creep. 

2// What do we learn from the strain of 17%. How good or bad these numbers are when compared to the existing alloys. 

3// What is the reason to opt for 790°C/700 MPa temperature-pressure combination. How does the alloy behave at other temperature-pressures?

4// It will be good to provide morphology in Figure 4 for completely deformed alloy. 

5// The results are promising but I am still confused with the overall objective of the work. The authors should compare their results with those from existing alloys/reported in the literature. Moreover, I am not sure about the reason for choosing temperature/pressure combination. May be the authors can include some text in introduction along with comparison with other systems in the results and discussion section, to inprovise the work.

Based on these observations, I recommend MAJOR revision for the present manuscript. 

Author Response

Response to Reviewer 1 Comments

Point 1:Figure 2 present alloy creep curve at 790°C/700 MPa. What is the reason to only choose 211 h for measurements and what happens after that measurement time (it will be good to extend the curve). Please also discuss reasons for primary creep and accelrated creep. 

Response 1: The fracture of the alloy occurs after 211h creep, so the creep curve ends at 211h. 

In 3.1 section, Lines 133 to 139 were added as:“this is because a large number of dislocations move in the matrix at the initial creep stage. As the creep time increased, the dislocation density of the alloy increased. Furthermore, its deformation-hardening effect caused the strain rate of the alloy to decrease. Meanwhile, thermal activation caused the dislocations in the alloy to slip and climb, which could release or slow the stress concentration in the local area, and the phenomenon of recovery softening occurred. When the deformation hardening and recovery softening reached a balance, the creep of the alloy entered a steady-state stage.” to discuss reasons for primary creep and accelrated creep.

Point 2: What do we learn from the strain of 17%. How good or bad these numbers are when compared to the existing alloys. 

Response 2: In 3.1 section, Line 146  was added as:“it shows that the alloy has good plasticity compared to similar alloys [26]”. Rreference 26 was added  in Reference section.

Point 3:  What is the reason to opt for 790°C/700 MPa temperature-pressure combination. How does the alloy behave at other temperature-pressures?

Response 3: In 2 section, Lines 76 to 78 were added as:“In the process of service, the aerospace engines experiences from medium temperature(700-900°C)/high stress to high temperature(900-1100°C)/low stress, aerospace engines blade must bear various conditions of starting to stabilization”. So opt for 790°C/700 MPa temperature-pressure combination. 

In 3.1 section, Lines 144 to 145 were added as:“The strain rates under steady-state creep at 780°C/700 MPa and 800°C/700 MPa were measured to be 0.008%/h, 0.039%/h , and the alloy’s creep life is 339 h, 133.4 h(Creep curve omitted )”.

Point 4:  It will be good to provide morphology in Figure 4 for completely deformed alloy. 

Response 4: Fig.4(b) is the morphology completely deformed alloy.

Point 5:  The results are promising but I am still confused with the overall objective of the work. The authors should compare their results with those from existing alloys/reported in the literature. Moreover, I am not sure about the reason for choosing temperature/pressure combination. May be the authors can include some text in introduction along with comparison with other systems in the results and discussion section, to inprovise the work.

Response 5: In 1 section, the last  paragraph was  modified as:“Based on this, the creep properties of DS Ni-based alloys were tested under different temperature conditions of 700MPa. Combined with microstructure observations and contrast analysis of the dislocation configuration, analyze the deformation mechanism of the medium temperature creep at different times and whether the alloy forms a K-W lock. And study the fracture characteristics of the alloy under the medium temperature creep, and provide a theoretical basis for the alloy design and manufacture”.

In the manuscript, all of the modified have been coated by the red color.

Reviewer 2 Report

The paper deals with the creep damage of a DS Ni-based superalloy. It represents a detailed microscopical analysis by SEM and TEM.

  • Please specify the exact number of specimens tested in section 2. It can be just guessed from the text of section 3 that they were 2? (up to rupture 211h and interrupted 150h test time).
  • It would be also interesting to present the interrupted test creep curve (150h) and how it matches with the curve up to rupture in Fig.2.
  • Please specify the microscopes types and manufacturers to section 2.
  • It would be useful to add the specimen drawing, not only the text description of the specimens. 
  • Fig. 3 especially e and f are not very sharp. If possible improve the quality.
  • The text font and its size are not the same throughout the paper.

The paper is otherwise clearly written and the analysis and conclusions are sound. I recommend its publishing after minor revision.

Author Response

Response to Reviewer 2 Comments

Point 1:  Please specify the exact number of specimens tested in section 2. It can be just guessed from the text of section 3 that they were 2? (up to rupture 211h and interrupted 150h test time).

Response 1: In 2 section, Lines 101 to 103 were added as: “Five specimens were used for creep test of 780℃/700 MPa, 790℃/ 700 MPa, 800℃/700 MPa and 790℃/700 MPa for 50 h and 150 h, respectively”. The exact number of specimens tested is five.

Point 2:It would be also interesting to present the interrupted test creep curve (150 h) and how it matches with the curve up to rupture in Fig.2.

Response 2: The interrupted test creep curve (150h) is similar to the creep curve up to rupture in Fig.2 from 0 h to 150 h, so it was omitted.

Point 3: Please specify the microscopes types and manufacturers to section 2.

Response 3: In 2 section,  “HITACHI” was added in line 97 as manufacturer of scanning electron microscopy. “Tecnai-20”and “ThermoFisher” were added in line 104 as type and manufacturer of transmission electron microscopy.

Point 4:  Fig. 3 especially e and f are not very sharp. If possible improve the quality.

Response 5: Adjusted the brightness and contrast of the picture in Fig. 3 to try to make the pictures clearer.

Point 5:  The text font and its size are not the same throughout the paper.

Response 6: Adjusted the text font and size throughout the paper.

In the manuscript, all of the modified have been coated by the red color.

Author Response

Response to Reviewer 3 Comments

Point a: In the title and in the body if you mention moderate temperature you need to specify for Ni-based superalloy what temperature range you consider as moderate? In this case, you only report results to a single temperature and stress level, you can simply mention that in your title.

Response a: In 2 section, Lines 76 to 78 were added as:“In the process of service, the aerospace engines experiences from medium temperature(700-900°C)/high stress to high temperature(900-1100°C)/low stress, aerospace engines blade must bear various conditions of starting to stabilization”.

Point b: Abstract: You mentioned multiple tests, but in the manuscript, you report only one test curve. There is some additional issues mentioned later.

Line 11: you mentioned “late stage of creep”, in the body you have mentioned the same with different terms such as “late stage of creep” and “accelerated creep”; avoid such confusion.

Line 19: you mentioned about creep resistance but did not connect in your result discussion. Response b: In 3.1 section, Lines 144 to 145 were added as:“The strain rates under steady-state creep at 780°C/700 MPa and 800°C/700 MPa were measured to be 0.008%/h, 0.039%/h , and the alloy’s creep life is 339 h, 133.4 h(Creep curve omitted )”.

Use “late stage of creep” to describe the “late stage of creep”.

In 4 section, 4.3 section was added to describe “Influence of deformation mechanism on creep resistance”.

Point c: Introduction: It should be defined what is considered in this paper as low temperature, moderate and high temperature range.

Line 57: There is several articles available on DS Ni based superalloy, not sure why you cite Rene80?

Line 58: consider revising the word “Crept” to “creep” here and entire body.

Line 61: Not sure what you mean by “same condition”.

Line 62: High temperature term is used, need to define what temperature range is considered as high here, see first comment.

Line 74-76: Need citation for this statement.

Line 79-80: It seems that there is literature available around the temperature reported in this manuscript for Ni based superalloy. Should consider cite those, and clearly state what new is add  ition in this paper. This is very important justifying novelty of this work. At this state, the work lacks novelty.

Response c: In 2 section, Lines 76 to 78 were added as:“In the process of service, the aerospace engines experiences from medium temperature(700-900°C)/high stress to high temperature(900-1100°C)/low stress, aerospace engines blade must bear various conditions of starting to stabilization”.

 Use DZ4 DS Ni based superalloy instead of  Rene80.

Revised the word “Crept” to “creep” entire body.

The “same condition” in Line 63 indicates that the temperature and stress during creep are the same.

In 2 section, Lines 76 to 78 were added as:“In the process of service, the aerospace engines experiences from medium temperature(700-900°C)/high stress to high temperature(900-1100°C)/low stress, aerospace engines blade must bear various conditions of starting to stabilization”.

Reference 23 was added for the statement “Line 74-76” in Reference section.

In 1 section, the last paragraph was modified as:“Based on this, the creep properties of DS Ni-based alloys were tested under different temperature conditions of 700MPa. Combined with microstructure observations and contrast analysis of the dislocation configuration, analyze the deformation mechanism of the medium temperatures creep at different times and whether the alloy forms a K-W lock. And study the fracture characteristics of the alloy under the medium temperature creep, and provide a theoretical basis for the alloy design and manufacture”.

Point d: Experimental procedure: Please consider including any test standard used for the study.

Response d: In 2 section,  the experimental procedure was introduced.

Point e: Line 127: different size letter/font, revise.

Response e: Unified letter size and font.

Point f:  Figure 2: Need more data.

Response f: The interrupted test creep curve (50 h and 150 h) is similar to the creep curve up to rupture in Fig.2 from 0 h to 50h and 150 h, so it was omitted.

Point g:  Line 144: unstressed might not be the right word, revise.

Response g: Use “minimum stress” instead of “unstressed”.

Point h:  Figure 4 and 8: Revise “Crept” with “Creep”

Response h: Revise “Crept” with “Creep”in  Fig. 4 and 8.

Point i:  Paragraph starting in line 192 and paragraph starting line 211 has different size/font, revise.

Response i: Unified letter size and font.

Point j:  Line 265: The equation should be in middle.

Response j: The equation has been placed in the middle.

Point k: Line 269 and Figure 7: It is reported that the Fig. 7(a) is of 150hrs, that means at the end of steady state creep and start of tertiary creep. However, you reported only one test until fracture. It is not clear how you took that image.

Response k: In 2 section, Lines 101 to 103 were added as: “Five specimens were used for creep test of 780℃/700 MPa, 790℃/ 700 MPa, 800℃/700 MPa and 790℃/700 MPa for 50 h and 150 h, respectively”. The exact number of specimens tested is five.

Point l: Line 302: It is mentioned different tests and different times are performed. Clarify what you mean by different time? Or until fracture? Report all data.

Response l: In 2 section, Lines 101 to 103 were added as: “Five specimens were used for creep test of 780℃/700 MPa, 790℃/ 700 MPa, 800℃/700 MPa and 790℃/700 MPa for 50 h and 150 h, respectively”. The exact number of specimens tested is five. The interrupted test creep curve (50 h and 150 h) is similar to the creep curve up to rupture in Fig.2 from 0 h to 50h and 150 h, so it was omitted.

In 3.1 section, Lines 144 to 145 were added as:“The strain rates under steady-state creep at 780°C/700 MPa and 800°C/700 MPa were measured to be 0.008%/h, 0.039%/h , and the alloy’s creep life is 339 h, 133.4 h(Creep curve omitted )”.

Point m: A model power law is stated (equation 15) that is generally used for steady state? Did not mention what model should be used for primary stage and tertiary stage?

Data fitting using the proposed model should be performed to validate the proposed constitutive model. Which is missing.

Response m: The strain rate of the alloy in the steady-state creep stage is the smallest, and the time in the steady-state creep stage is longer than that in the first and third creep stages. Therefore, the creep life of the alloy is determined by the creep steady-state life. So, in this paper, the constitutive equation of creep induced by local stress (sG) of gmatrix during steady-state creep is established. Because the dislocation movement is violent in the first and third stages of creep, the alloy deformation rate is relatively large, and the duration is relatively short. Therefore, it is not easy to construct the constitutive equation at this point. Generally, the influencing factors are determined by observing the creep curve and the experimental method of the structure morphology.

The constitutive model constructed in this paper is related to many variables, so it is difficult to use experimental methods to fit the model to the data.

Point n:  Equation 16: put it in middle.

Response n: The equation has been placed in the middle.

Point o: Different font/size used in 2nd and third paragraph. Please proofread before submission.

Response o:Unified letter size and font.

Point p:  Consider citing English articles, some articles are in Chinese cannot be verified. Reference [14] missing title of the paper. Ref 17-23 remove underline.

Response p: The above references have been modified. 

In the manuscript, all of the modified have been coated by the red color.

Round 2

Reviewer 1 Report

The manuscript is revised as per suggestions. It can be accepted for publication.

Author Response

Thank you for agreeing to publish our article.